# Scaling & Shifting Your Features: A New Baseline for Efficient Model Tuning

**Dongze Lian**[1*]   **Daquan Zhou**[1,2*]   **Jiashi Feng**[2]   **Xinchao Wang**[1]
[1]National University of Singapore    [2]ByteDance
{dongze,xinchao}@nus.edu.sg   {zhoudaquan21,jshfeng}@gmail.com

## Abstract

Existing fine-tuning methods either tune all parameters of the pre-trained model (full fine-tuning), which is not efficient, or only tune the last linear layer (linear probing), which suffers a significant accuracy drop compared to the full fine-tuning. In this paper, we propose a new parameter-efficient fine-tuning method termed as SSF, representing that researchers only need to Scale and Shift the deep Features extracted by a pre-trained model to catch up with the performance of full fine-tuning. In this way, SSF also surprisingly outperforms other parameter-efficient fine-tuning approaches even with a smaller number of tunable parameters. Furthermore, different from some existing parameter-efficient fine-tuning methods (*e.g.*, Adapter or VPT) that introduce the extra parameters and computational cost in the training and inference stages, SSF only adds learnable parameters during the training stage, and these additional parameters can be merged into the original pre-trained model weights via re-parameterization in the inference phase. With the proposed SSF, our model obtains 2.46% (90.72% *vs.* 88.54%) and 11.48% (73.10% *vs.* 65.57%) performance improvement on FGVC and VTAB-1k in terms of Top-1 accuracy compared to the full fine-tuning but only fine-tuning about 0.3M parameters. We also conduct amounts of experiments in various model families (CNNs, Transformers, and MLPs) and datasets. Results on 26 image classification datasets in total and 3 robustness & out-of-distribution datasets show the effectiveness of SSF. Code is available at https://github.com/dongzelian/SSF.

## 1 Introduction

With the popularity of the data-driven methods in the deep learning community, the dataset scale and the model size have both got huge explosions. There is a tendency to explore large models and then adopt these pre-trained models in downstream tasks to achieve better performance and faster convergence, which gradually becomes a common way.

However, the current procedure depends on full fine-tuning heavily, where all the parameters of the model are updated. It inevitably causes the model to be over-fitted to the small target dataset and thus cannot be used for other tasks after the fine-tuning. As a result, the device will need to save a dedicated set of model parameters for each task, which causes a huge amount of storage space, especially for today's large models (*e.g.*, ViT-G/14 [11] 1.8G, CoAtNet [5] 2.4G).

A simple solution for the above problem is linear probing [16], where only the last head layer is fine-tuned. However, this practice usually yields inferior performance compared to the full fine-tuning proxy. Motivated by the success of the parameter-efficient fine-tuning strategy with prompt in the field of natural language processing (NLP) [21, 32, 23, 19], the recent work implements a similar proxy on vision tasks [29], termed as Visual Prompt Tuning (VPT). Specifically, VPT [29] proposes to insert learnable prompts as inputs and append them to the original image tokens. These prompts

---

*Equal contribution.

36th Conference on Neural Information Processing Systems (NeurIPS 2022).

| Method | Acc. | Params. (M) | Unified parameter space | No extra inference params. |
|---|---|---|---|---|
| Full fine-tuning | 93.82 | 85.88 | ✓ | ✓ |
| Linear probing | 88.70 | 0.08 | ✓ | ✓ |
| Adapter [21] | 93.34 | 0.31 | ✓ | ✗ |
| VPT [29] | 93.17 | 0.54 | ✗ | ✗ |
| SSF (ours) | **93.99** | 0.28 | ✓ | ✓ |

Table 1: Characteristics of different fine-tuning methods. Acc. means the Top-1 accuracy (%) on CIFAR-100 with a pre-trained ViT-B/16 for tuning. Params. means the learnable parameters at fine-tuning. Our SSF has a unified learnable parameter space and does not require extra inference parameters while obtaining superior performance.

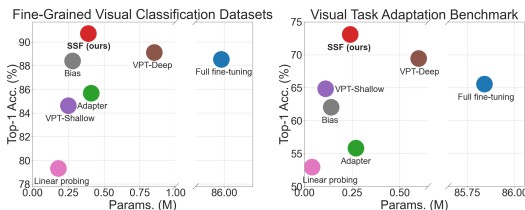

Figure 1: Performance comparisons of seven fine-tuning methods with a pre-trained ViT-B/16 model on the FGVC dataset and VTAB-1k benchmark. Our SSF (red dots) achieves state-of-the-art performance only with about 0.3M average learnable parameters.

will interact with the image tokens by performing self-attention and are updated during the fine-tuning process. In this manner, a significant performance improvement can be achieved in downstream tasks compared to a linear probing proxy. Nevertheless, compared to the full fine-tuning and linear probing, it additionally raises two issues: i) VPT tunes the number of prompts for different tasks, which introduces a task-dependent learnable parameter space. The fine-tuning performance is sensitive to the number of prompts for each task and needs to be carefully designed. Too few or too many prompts might either degrade the accuracy of fine-tuning or increase the redundancy of the computation (*e.g.*, 200 prompts on Clevr/count *vs.* 1 prompt on Flowers102); ii) VPT [29], as well as other Adapter-based methods [21, 42], introduces additional parameters and computational cost in the inference phase compared to the original pre-trained model. For instance, VPT introduces additional inputs for self-attention with image tokens. Adapter-based methods insert additional modules into the pre-trained model. These methods change the specific backbone architecture or the input of the network, which might result in frequent structure modifications and heavy workload, especially for those models that are already deployed in edge devices (*e.g.*, mobile phones).

To cope with the above issues, we attempt to find a general proxy for parameter-efficient fine-tuning, where the learnable parameter space is unified (task-independent) and no additional inference parameters are introduced. Inspired by some feature modulation methods [58, 25, 45], we propose a new parameter-efficient fine-tuning method named SSF, where you only need to Scale and Shift your deep Features extracted by a pre-trained model for fine-tuning. The intuition behind our approach come from the fact that the upstream datasets and downstream datasets have different data distributions [50]. Therefore, it is difficult to apply the model weights trained in the upstream dataset to the downstream dataset. For instance, a naive linear probing strategy with keeping the weights of backbone frozen will cause performance degradation. To alleviate the above problem, SSF introduces scale parameters and shift parameters, which could be considered as variance and mean to modulate the features of the downstream dataset extracted with the pre-trained model on the upstream dataset, such that the modulated feature falls in a discriminative space. These scale parameters and shift parameters do not depend on any input and have a unified learnable parameter space for different tasks. Another advantage of SSF is that it only introduces linear transformations because we scale and shift the extracted features. These linear transformations could be further merged into the original pre-trained weight via model re-parameterization [10] in the inference phase, thus avoiding the extra parameters and FLOPs for downstream tasks. For a deployed model in edge devices, only the updated weights after fine-tuning need to be uploaded instead of changing the backbone architecture. Table 1 shows the specific characteristics comparisons between SSF and other fine-tuning methods. SSF is simple, effective, and efficient, which also conforms to Occam's Razor principle. Therefore, we explore this new baseline and find that it surprisingly outperforms all other parameter-efficient fine-tuning methods.

We evaluate our method on 26 classification datasets in total and 3 robustness & out-of-distribution datasets. SSF obtains state-of-the-art performance compared to other parameter-efficient fine-tuning methods with the trainable parameters and accuracy trade-off (Table 1 and Figure 1). Compared to the full fine-tuning, our method obtains 2.46% (90.72% *vs.* 88.54%) and 11.48% (73.10% *vs.* 65.57%) performance improvement on FGVC and VTAB-1k in terms of Top-1 accuracy but only with about 0.3M trainable parameters. Furthermore, our SSF does not require additional parameters during the inference phase. It is plug-and-play and is very easy to extend to various model families (CNNs, Transformers, and MLPs). Our SSF establishes a new baseline and we hope that it brings more insight into the field of the efficient model tuning.

## 2 Related Work

### 2.1 Model Families

Convolution has been used for a long time as the main module to extract the image features in computer vision tasks, and CNN-based architectures have been studied [49, 18, 59, 39, 60, 37, 63] with extension on graph-based data [62, 61, 36]. Recently, another architecture family, Transformer, has gained widespread attention owing to its great success in NLP [56, 8, 23]. Following this direction, Dosovitskiy *et al.* [11] first employ a transformer in the domain of computer vision and introduce a new architecture paradigm, ViT, which achieves promising results [64, 48]. Subsequently, various transformer-based models, such as DeiT [53] and Swin Transformer [38], are introduced and shown to be effective on a variety of tasks such as object detection, semantic segmentation, action recognition [40], *etc*. In another line, Tolstikhin *et al.* [52] propose a pure MLP-based architecture, and subsequent papers [20, 33] have interestingly demonstrated that the MLP-based architectures can catch up to transformers. However, in addition to the well-designed modules, their excellent performance is also attributed to the deployment of large-scale models. Given a large-scale model pre-trained on a large dataset, how to perform parameter-efficient fine-tuning in downstream tasks is essential but is currently less explored. In this paper, we propose SSF as a new baseline and show its promising performance with comprehensive validation in a wide variety of tasks.

### 2.2 Pre-training and Fine-tuning

Early models [18, 24, 22, 59, 51] are usually pre-trained on the ImageNet-1K dataset, and then fine-tuned on downstream tasks to achieve faster convergence [17] or better performance. Such a procedure is called pre-training and fine-tuning, or transfer learning. Recent works tend to employ larger models (*e.g.*, ViT [11] and Swin Transformer V2 [38]) and train them on larger datasets (*e.g.*, ImageNet-21K and JFT-300M) in pursuit of better performance. Both in the domains of NLP and computer vision, these large models [8, 38, 47, 15, 69, 70] achieve enormous performance improvements compared to the small-scale models and provide pre-trained weights for downstream tasks. Some other works attempt to explore how to efficiently fine-tune the pre-trained models [13, 71] on the target tasks. For instance, given a target task, SpotTune [13] investigates which layers need to be fine-tuned. Touvron *et al.* [54] find that fine-tuning the weights of the attention layers and freezing weights of the other parts is sufficient to adapt the vision transformers to other downstream tasks. Some works also propose to insert adapters into the network to fine-tune in a parameter-efficient way. These adapters can be a small non-linear network [21], a hyper-network that generates model weights [43], or a compactor [42] which performs a low-rank decomposition to reduce the parameters. Some works have also tried to only update the bias term [2, 66]. More recently, VPT [29] proposes to insert a small number of learnable parameters (prompts) and optimize them while freezing the backbone, which achieves significant performance improvement compared to the full fine-tuning. During the submission of this work, some methods [3, 68] are also proposed for parameter-efficient fine-tuning, *e.g.*, inserting a adapter module or neural prompt search. Different from all the above works, we propose to scale and shift deep features extracted by a pre-trained model, which is simple but effective and outperforms other parameter-efficient fine-tuning methods.

### 2.3 Feature Modulation

Many works have attempted to modulate features to obtain better performance. The most relevant ones to our work are various normalization methods [26, 1, 58]. BN, LN, and GN usually normalize the features and then transform them linearly with scale and shift factors to modulate feature distribution, which has been verified to be effective in amounts of tasks. STN [28] introduces a learnable module to spatially transform feature maps. In the field of image generation, AdaIN [25] generates scale and shift factors to characterize specific image styles. Self-modulation [4] shows GANs benefit from self-modulation layers in the generator. In vision-language tasks, Conditional BN [6] and FiLM [45] are often utilized to modulate the features of two modalities. Unlike some algorithms such as BN, our SSF is not limited to the modulation of normalization layer, and it has a different motivation that is to alleviate the distribution mismatch between upstream tasks and downstream tasks for parameter-efficient fine-tuning. As a comparison, we also conduct experiments in Sec. 4.3 and show that our SSF is more effective compared to only tuning the normalization layer. Compared to STN, AdaIN, FiLM and so on, our method is input-independent and these scale and shift parameters model the distribution of the whole dataset so that they can be absorbed into the original pre-trained model weights in the inference phase.

## 2.4  Model Re-parameterization

Model re-parameterization has been a common practice to improve inference efficiency. One of the representative techniques is batch normalization folding used in the model compression algorithms [27]. The parameters introduced by the batch normalization layers [26] are merged into the convolutional layers usually stacked before them. This technique is further utilized to merge different branches of networks into a new branch [65, 10, 9]. Similarly, our SSF fully adopts linear transformations, which allows the scale and shift parameters in the training phase to be merged into the original pre-trained model weights, thus avoiding the introduction of the extra parameters and computational cost during the inference phase.

# 3  Approach

## 3.1  Preliminaries

**Transformers.** In a vision transformer (ViT) [11], an RGB image $I \in \mathbb{R}^{3 \times H \times W}$ is divided into $N \times N$ non-overlapping patches, and then these image patches appended a class token are fed into an embedding layer followed by the $L$-layer vision transformer blocks with self-attention as the core operation. The input $x \in \mathbb{R}^{(N^2+1) \times d}$, where $d$ is the embedding dimension, is first transformed to keys $K \in \mathbb{R}^{(N^2+1) \times d}$, values $V \in \mathbb{R}^{(N^2+1) \times d}$, and queries $Q \in \mathbb{R}^{(N^2+1) \times d}$. After that, we can calculate a global self-attention by

$$\text{Attention}(Q, K, V) = \text{Softmax}(\frac{QK^T}{\sqrt{d}})V. \tag{1}$$

The output of the attention layer will be fed to a two-layer MLP to extract information in the channel dimension.

**Adapter.** Adapter [21] is inserted into the transformer layer for efficient fine-tuning. It is a bottleneck module with a few trainable parameters, which contains a down-projection to reduce the feature dimension, a non-linear activation function, and an up-projection to project back to the original dimension. Therefore, given the input $x \in \mathbb{R}^{(N^2+1) \times d}$, the output is calculated by

$$\text{out} = [W^{\text{up}}\phi(W^{\text{down}}x^T)]^T, \tag{2}$$

where $W^{\text{down}} \in \mathbb{R}^{d' \times d}$ (where $d' \ll d$), $\phi$, and $W^{\text{up}} \in \mathbb{R}^{d \times d'}$ represent the down-projection matrix, non-linear function, and up-projection matrix, respectively.

**VPT.** VPT [29] inserts some learnable parameters (*i.e.*, prompts) into the input space after the embedding layer. These prompts interact with the original image tokens by performing self-attention. During the fine-tuning, the weights of the backbone network are kept frozen and only the parameters of the prompts are updated. VPT-Shallow inserts prompts in the first layer while VPT-Deep inserts prompts in all the layers of the transformer. Assuming that the input is $x \in \mathbb{R}^{(N^2+1) \times d}$, denote the inserted prompts as $p \in \mathbb{R}^{n \times d}$, where $n$ is the number of prompts, the combined tokens $x'$ is

$$x' = [x; p], \tag{3}$$

where $x' \in \mathbb{R}^{(N^2+n+1) \times d}$ will be fed into the transformer block for self-attention (Eq. (1)).

## 3.2  Scaling and Shifting Your Features for Fine-tuning

Different from the above methods, we introduce both the scale and shift factors to modulate deep features extracted by a pre-trained model with linear transformation to match the distribution of a target dataset, as mentioned in Sec. 1. Five main properties are covered in our method: i) SSF achieves on-par performance with the full fine-tuning strategy; ii) all downstream tasks can be inputted to the model independently without relying on any other task; iii) the model only needs to fine-tune very few parameters; iv) unlike VPT [29], which adjusts the number of prompts for each task, the set of parameters for fine-tuning in SSF does not change as the task changes, making it feasible to further fine-tune the parameters later by adding more tasks for multi-task learning or continuous learning[2]; v) thanks to the linear transformation, SSF avoids the introduction of the extra parameters and computational cost during the inference phase, making our method zero overhead.

---

[2]It provides more flexibility, which is not a contradiction to ii).

**The design of SSF.** SSF performs the linear transformation to modulate the features for parameter-efficient fine-tuning as shown in Figure 2. In Figure 2 (a), given a model pre-trained in the upstream task, we insert SSF-ADA[3] after each operation (OP) of the network to modulate features. There are $K$ OPs in total and these operations might contain multi-head self-attention (MSA), MLP and layer normalization (LN), *etc*. During the fine-tuning, the pre-trained weights in these operations are kept frozen and the SSF-ADA parameters are kept updated. The specific SSF-ADA structure is shown in Figure2 (c), where the features output from the previous operation are performed dot product with a scale factor and then summed with a shift factor, which are input-independent. Formally, given the input $x \in \mathbb{R}^{(N^2+1) \times d}$, the output $y \in \mathbb{R}^{(N^2+1) \times d}$ (is also the input of the next operation) is calculated by

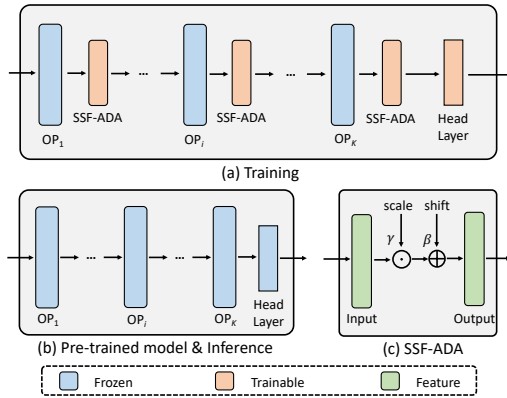

Figure 2: The overall pipeline of SSF. (a) Training pipeline via SSF, where an OP means an operation, *e.g.*, MSA, MLP or LN. (b) A pre-trained model or inference pipeline. (c) Our SSF-ADA.

$$y = \gamma \odot x + \beta, \tag{4}$$

where $\gamma \in \mathbb{R}^d$ and $\beta \in \mathbb{R}^d$ are the scale and shift factors, respectively. $\odot$ is the dot product.

**Re-parameterization.** Since SSF-ADA is a completely linear transformation, we can re-parameterize it by absorbing the scale and shift terms into the previous linear layer as follows

$$y = \gamma \odot x + \beta = \gamma \odot (w * t + b) + \beta = (\gamma \odot w) * t + \gamma \odot b + \beta, \tag{5}$$

where $w$ and $b$ are the weight and bias terms, respectively. $*$ represents the 'convolution' operation in the convolutional layer or the 'multiplication' operation in the MLP layer. $t$ is the input of the previous linear layer. Since $w$ and $b$ are frozen and $\gamma$ and $\beta$ are updated in the fine-tuning, $\gamma$ and $\beta$ can be merged into the original parameter space ($w$ and $b$) in the inference stage through the above formulation. From this perspective, our SSF-ADA makes it possible to perform downstream tasks without adding any extra parameters and computational costs, as shown in Figure2 (b).

**Discussion.** The first question is why we want the input $\gamma$ and $\beta$ to be input-independent. As FiLM [45] and AdaIN [25] show, we could obtain $\gamma$ and $\beta$ by conditioning an image sample, however, this might cause two shortcomings. One is that we want $\gamma$ and $\beta$ to be input-independent to represent the distribution of the whole downstream dataset so that we can modify the previous weight distribution to fit the downstream dataset by modulating the feature. Secondly, the conditional input requires the introduction of some additional networks (*e.g.*, MLPs) to generate $\gamma$ and $\beta$, which introduces more trainable parameters. More importantly, to better generate $\gamma$ and $\beta$, a non-linear activation function might be required, which will lead to the intractability of the re-parameterization. Therefore, we directly perform a fully linear transformation to merge the $\gamma$ and $\beta$ factors into the original pre-trained weights, so that weights can be easily uploaded to the edge devices without any modification of the backbone architecture.

The second question is which operations should be followed by SSF-ADA. Our experience is that you can insert SSF-ADA after each operation with a linear coefficient in ViT. Although we can search for some optimal layers or operations with Neural Architecture Search (NAS) [46, 35, 14, 34], to reduce the number of the trainable parameters, we believe that our method will produce better results (or not worse than NAS) without introducing too many trainable parameters that can be merged for inference, as will be shown in Sec. 4.3.

### 3.3 Complexity Analysis

We also compare the complexity of Adapter, VPT and our SSF. Take a ViT as an example, the dimension and number of the tokens are $d$ and $N^2$. Assuming that Adapter projects features from $d$-dim to $d'$-dim (where $d' \ll d$) so that the extra trainable parameters are $2dd'$ in each layer,

---

[3]Here, we refer to our proposed method as SSF and the specific module as SSF-ADA.

VPT inserts $n$ prompts to obtain $nd$ extra parameters in each layer, and SSF inserts SSF-ADA after each operation with a linear coefficient to obtain $md$ extra parameters in each layer, when the total number of layers is $L$, the complexity of Adapter, VPT and SSF is shown in Table 2. The specific number of additional parameters used by

| Method | Adapter | VPT-Shallow | VPT-Deep | SSF (ours) |
|---|---|---|---|---|
| # Extra Params. | $2Ldd'$ (1) | $nd$ (1) | $nLd$ (1) | $mLd$ (0) |
| # Extra FLOPs | $2N^2Ldd'$ (1) | $2n(2N^2+n)d$ (1) | $2n(2N^2+n)Ld$ (1) | $mN^2Ld$ (0) |

Table 2: The complexity comparisons of Adapter [21], VPT [29] and our SSF. '(1)': the same parameters and FLOPs for training and inference; '(0)': no additional parameters and FLOPs are required for inference.

Adapter, VPT and SSF depends on the values of $d'$, $n$ and $m$. However, in practice, SSF outperforms Adapter and VPT-Deep even with slightly fewer parameters in the training stage as we will see in Sec. 4. Further, in the inference stage, borrowing the model re-parameterization strategy, the extra parameters and FLOPs of SSF are zero. However, the complexity of Adapter and VPT remain the same compared to the training, which establishes the strengths of our approach.

## 4 Experiments

### 4.1 Experimental Settings

**Datasets**. We mainly conduct our experiments on a series of datasets that can be categorized into three types as detailed below:

*FGVC*. Following VPT [29], we employ five Fine-Grained Visual Classification (FGVC) datasets to evaluate the effectiveness of our proposed SSF, which consists of CUB-200-2011 [57], NABirds [55], Oxford Flowers [44], Stanford Dogs [30] and Stanford Cars [12].

*VTAB-1k*. VTAB-1k benchmark is introduced in [67], which contains 19 tasks from diverse domains: i) Natural images that are captured by standard cameras; ii) Specialized images that are captured by non-standard cameras, *e.g.*, remote sensing and medical cameras; iii) Structured images that are synthesized from simulated environments. This benchmark contains a variety of tasks (*e.g.*, object counting, depth estimation) from different image domains and each task only contains 1,000 training samples, thus is extremely challenging.

*General Image Classification Datasets*. We also validate the effectiveness of SSF on general image classification tasks. We choose the CIFAR-100 [31] and ImageNet-1K [7] datasets as evaluation datasets, where CIFAR-100 contains 60,000 images with 100 categories. ImageNet-1K contains 1.28M training images and 50K validation images with 1,000 categories, which are very large datasets for object recognition.

**Models**. For a fair comparison, we follow VPT [29] and mainly select ViT-B/16 [11] model pre-trained on ImageNet-21K as the initialization for fine-tuning. In addition, we also generalize our method to backbones of different model families, including the recent Swin Transformer [38] (Swin-B), ConvNeXt-B [39] and AS-MLP-B [33]. The former builds a hierarchical transformer-based architecture, and the latter two belong to CNN-based architecture and MLP-based architecture respectively.

**Baselines**. We first compare our method with the two basic fine-tuning methods: i) full fine-tuning, where all parameters of the models are updated at fine-tuning; ii) linear probing, where only the parameters of the classification head (an MLP layer) are updated. We also compare our method with recent parameter-efficient fine-tuning methods: iii) Adapter [21], where a new adapter structure with up-projection, non-linear function, and down-projection is inserted into the transformer and only the parameters of this new module are updated; iv) Bias [66], where all the bias terms of parameters are updated; v) VPT [29], where the prompts are inserted into transformers as the input tokens and they are updated at fine-tuning.

**Implementation Details.** For the FGVC datasets, we process the image with a randomly resize crop to $224 \times 224$ and a random horizontal flip for data augmentation. For VTAB-1k, we directly resize the image to $224 \times 224$, following the default settings in VTAB [67]. For CIFAR-100 and ImageNet-1K, we follow the fine-tuning setting of ViT-B/16 in [11], where the stronger data augmentation strategies are adopted. We employ the AdamW [41] optimizer to fine-tune models for 100 epochs for CIFAR-100, and 30 epochs for ImageNet-1K. The cosine decay strategy is adopted for the learning rate schedule, and the linear warm-up is used in the first 10 epochs for CIFAR-100 and 5 epochs for ImageNet-1K.

| Method \ Dataset | CUB-200-2011 | NABirds | Oxford Flowers | Stanford Dogs | Stanford Cars | Mean | Params. (M) |
|---|---|---|---|---|---|---|---|
| Full fine-tuning | 87.3 | 82.7 | 98.8 | 89.4 | 84.5 | 88.54 | 85.98 |
| Linear probing | 85.3 | 75.9 | 97.9 | 86.2 | 51.3 | 79.32 | 0.18 |
| Adapter [21] | 87.1 | 84.3 | 98.5 | 89.8 | 68.6 | 85.67 | 0.41 |
| Bias [66] | 88.4 | 84.2 | 98.8 | **91.2** | 79.4 | 88.41 | 0.28 |
| VPT-Shallow [29] | 86.7 | 78.8 | 98.4 | 90.7 | 68.7 | 84.62 | 0.25 |
| VPT-Deep [29] | 88.5 | 84.2 | 99.0 | 90.2 | 83.6 | 89.11 | 0.85 |
| SSF (**ours**) | **89.5** | **85.7** | **99.6** | 89.6 | **89.2** | **90.72** | 0.39 |

Table 3: Performance comparisons on five FGVC datasets with ViT-B/16 models pre-trained on ImageNet-21K.

| | Natural | | | | | | | Specialized | | | | Structured | | | | | | | | | |
|---|---|---|---|---|---|---|---|---|---|---|---|---|---|---|---|---|---|---|---|---|---|
| Method \ Dataset | CIFAR-100 | Caltech101 | DTD | Flowers102 | Pets | SVHN | Sun397 | Patch Camelyon | EuroSAT | Resisc45 | Retinopathy | Clevr/count | Clevr/distance | DMLab | KITTI/distance | dSprites/loc | dSprites/ori | SmallNORB/azi | SmallNORB/ele | Mean | Params. (M) |
| Full fine-tuning [29] | 68.9 | 87.7 | 64.3 | 97.2 | 86.9 | 87.4 | 38.8 | 79.7 | 95.7 | 84.2 | 73.9 | 56.3 | 58.6 | 41.7 | 65.5 | 57.5 | 46.7 | 25.7 | 29.1 | 65.57 | 85.84 |
| Linear probing [29] | 63.4 | 85.0 | 63.2 | 97.0 | 86.3 | 36.6 | 51.0 | 78.5 | 87.5 | 68.6 | 74.0 | 34.3 | 30.6 | 33.2 | 55.4 | 12.5 | 20.0 | 9.6 | 19.2 | 52.94 | 0.04 |
| Adapter [21] | 74.1 | 86.1 | 63.2 | 97.7 | 87.0 | 34.6 | 50.8 | 76.3 | 88.0 | 73.1 | 70.5 | 45.7 | 37.4 | 31.2 | 53.2 | 30.3 | 25.4 | 13.8 | 22.1 | 55.82 | 0.27 |
| Bias [66] | 72.8 | 87.0 | 59.2 | 97.5 | 85.3 | 59.9 | 51.4 | 78.7 | 91.6 | 72.9 | 69.8 | 61.5 | 55.6 | 32.4 | 55.9 | 66.6 | 40.0 | 15.7 | 25.1 | 62.05 | 0.14 |
| VPT-Shallow [29] | 77.7 | 86.9 | 62.6 | 97.5 | 87.3 | 74.5 | 51.2 | 78.2 | 92.0 | 75.6 | 72.9 | 50.5 | 58.6 | 40.5 | 67.1 | 68.7 | 36.1 | 20.2 | 34.1 | 64.85 | 0.11 |
| VPT-Deep [29] | **78.8** | 90.8 | 65.8 | 98.0 | 88.3 | 78.1 | 49.6 | 81.8 | **96.1** | 83.4 | 68.4 | 68.5 | 60.0 | 46.5 | 72.8 | 73.6 | 47.9 | **32.9** | 37.8 | 69.43 | 0.60 |
| SSF (**ours**) | 69.0 | **92.6** | **75.1** | **99.4** | **91.8** | **90.2** | **52.9** | **87.4** | 95.9 | **87.4** | **75.5** | **75.9** | **62.3** | **53.3** | **80.6** | **77.3** | **54.9** | 29.5 | **37.9** | **73.10** | 0.24 |

Table 4: Performance comparisons on the VTAB-1k benchmark with ViT-B/16 models pre-trained on ImageNet-21K.

## 4.2 Performance Comparisons on Image Classification

We compare the performance of our SSF and other baseline methods in 26 image classification tasks and the results on FGVC and VTAB-1k are shown in Table 3 and Table 4 (also see Figure 1), respectively, and the results on CIFAR-100 and ImageNet-1K are shown in Table 5, which are evaluated in Top-1 accuracy (%). In these three tables, the bold font shows the best accuracy of all methods and the underline font shows the second best accuracy.

We have the following findings by observing them: i) In Table 3 and Table 4, where the last column is the average of the fine-tuned parameters for each method on the corresponding datasets, our SSF outperforms VPT [29] and other parameter-efficient fine-tuning methods, and even achieves better performance than full fine-tuning, which is mainly owing to the linear transformation applied on the features. Specifically, SSF obtains 1.81% (90.72% *vs.* 89.11%) and 2.46% (90.72% *vs.* 88.54%) accuracy improvement on five FGVC datasets, and 5.29% (73.10% *vs.* 69.43%) and 11.48% (73.10% *vs.* 65.57%) improvement on the VTAB-1k benchmark compared to VPT and full fine-tuning. Meanwhile, SSF also uses fewer trainable parameters compared to VPT-Deep in both datasets (0.39M *vs.* 0.85M, 0.24M *vs.* 0.60M). SSF maintains a unified learnable parameter space for different tasks with a few parameters while VPT [29] needs to design the different number of prompts for each task, which also shows the conciseness of our approach; ii) In Table 5, *i.e.*, in CIFAR-100 and ImageNet-1K, SSF and other parameter-efficient fine-tuning methods have difficulty in achieving the similar performance to the full fine-tuning, probably because these datasets have sufficient data to prevent over-fitting of the model, especially in ImageNet-1K. In contrast, in the VTAB-1k benchmark, the amount of data is not very large (*e.g.*, only 1,000 training images), which might cause over-fitting of the model for the full fine-tuning. Nevertheless, in CIFAR-100 and ImageNet-1K, our SSF still outperforms previous parameter-efficient fine-tuning methods (Adapter, Bias, and VPT), which shows the effectiveness of our method; iii) In Table 5, the results of our SSF with Swin Transformer, ConvNeXt, and AS-MLP models consistently outperform those of other parameter-efficient fine-tuning methods, which also verifies the effectiveness of SSF on a wide variety of models.

**Computational cost.** To validate the efficiency of our method, we show the computational cost of SSF in Figure 3. We employ a batch size of 16 for the training stage and inference stage, and use mixed precision training. All running results in Figure 3 are measured in a single GeForce RTX 2080Ti GPU. We can see that SSF has similar training time and training memory with VPT but with

| Model | ViT-B/16 [11] | | | | Swin-B [38] | | | | ConvNeXt-B [39] | | | | AS-MLP-B [33] | |
|---|---|---|---|---|---|---|---|---|---|---|---|---|---|---|
| Dataset / Method | CIFAR-100 | Params. (M) | ImageNet-1K | Params. (M) | CIFAR-100 | Params. (M) | ImageNet-1K | Params. (M) | CIFAR-100 | Params. (M) | ImageNet-1K | Params. (M) | CIFAR-100 | Params. (M) |
| Full fine-tuning | 93.82 | 85.88 | **83.58** | 86.57 | **93.85** | 86.85 | **85.20** | 88.03 | **94.14** | 87.67 | **85.80** | 88.85 | **89.96** | 86.83 |
| Linear probing | 88.70 | 0.08 | 82.04 | 0.77 | 89.27 | 0.10 | 83.25 | 1.03 | 89.20 | 0.10 | 84.05 | 1.03 | 79.04 | 0.10 |
| Adapter [21] | 93.34 | 0.31 | 82.72 | 1.00 | 92.49 | 0.33 | 83.82 | 1.26 | 92.86 | 0.45 | 84.49 | 1.37 | 88.01 | 0.33 |
| Bias [66] | 93.39 | 0.18 | 82.74 | 0.87 | 92.19 | 0.24 | 83.92 | 1.16 | 92.80 | 0.23 | 84.63 | 1.16 | 87.46 | 0.26 |
| VPT-Shallow [29] | 90.38 | 0.23 | 82.08 | 0.92 | 90.02 | 0.13 | 83.29 | 1.05 | - | - | - | - | - | - |
| VPT-Deep [29] | 93.17 | 0.54 | 82.45 | 1.23 | 92.62 | 0.70 | 83.44 | 1.63 | - | - | - | - | - | - |
| SSF **(ours)** | **93.99** | 0.28 | 83.10 | 0.97 | 93.06 | 0.37 | 84.40 | 1.29 | 93.45 | 0.36 | 84.85 | 1.28 | 88.28 | 0.37 |

Table 5: Performance comparisons on CIFAR-100 and ImageNet-1K with various model families, where ViT-B/16, Swin-B, and ConvNeXt-B are pre-trained on ImageNet-21K, and AS-MLP-B is pre-trained on ImageNet-1K.

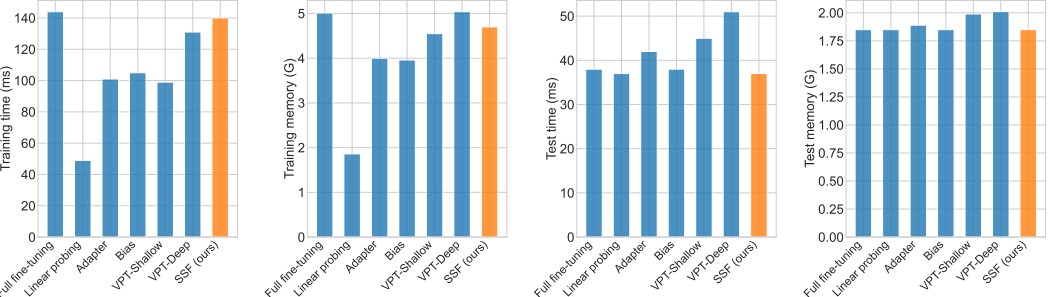

Figure 3: Computational cost of different tuning methods. From left to right: training time, training memory, test time, and test memory.

less inference time and inference memory. Here, we show the computational cost of VPT with 200/50 prompts (the same number of prompts to obtain the performance in Table 5) for VPT-Shallow and VPT-Deep, respectively. When adding the number of prompts, the time cost and memory will be larger but our SSF achieves zero-overhead inference, which is more advantageous.

### 4.3 The Impacts of Different Designs

As the core operation of SSF, we thoroughly evaluate how SSF-ADA affects results, *e.g.*, the insertion locations, the initialization of SSF-ADA and its components. We conduct experiments to analyze the impacts of different designs for fine-tuning. All experiments are implemented with pre-trained ViT-B/16 models on CIFAR-100 and the results are shown in Table 6.

**The impact of the number of layers.** We directly insert SSF-ADA into different layers to evaluate the effect of inserting layers, and the results are shown in table 6a. The values in the #layers column indicate the number of layers with SSF-ADA, where #layers-0 represents linear probing. From the first and second rows, we find that the results will improve from 88.70% to 92.69% and grow with a small number of trainable parameters (0.08M *vs.* 0.11M) when only inserting SSF-ADA into the first two layers. Keep adding SSF-ADA in the subsequent layers will make the results better. The growth of the results is almost linear with the number of layers of inserted SSF-ADA. Therefore, we directly choose to insert SSF-ADA into all (12) layers of vision transformer to bring the best results (93.99%) with 0.28M trainable parameters.

**The impact of the different insertion locations.** Based on the different operations of ViT, we evaluate the impact of the insertion locations of SSF-ADA. We separately remove SSF-ADA after these operations and the results are shown in Table 6b. We find that removing the SSF-ADA in the MLP operation achieves inferior results than removing those in the Attention operation (93.46% *vs.* 93.69%) with comparable trainable parameters (0.19M *vs.* 0.21M), which suggests that performing feature modulation for the MLP operation might be more important. Although one can use NAS to search for the importance of different operations and thereby insert SSF-ADA in specific locations, the results might not be better than inserting SSF-ADA in all operations. Therefore, in order to obtain excellent performance, we do not perform NAS but directly insert SSF-ADA into all operations.

| #layers | Acc. | Params. |
|---|---|---|
| 0 | 88.70 | 0.08 |
| 2 | 92.69 | 0.11 |
| 4 | 93.30 | 0.15 |
| 8 | 93.60 | 0.22 |
| 12 (ours) | **93.99** | 0.28 |

(a)

| location | Acc. | Params. |
|---|---|---|
| w/o. mlp | 93.46 | 0.19 |
| w/o. attn | 93.69 | 0.21 |
| w/o. embed | 93.91 | 0.28 |
| w/o. norm | 93.80 | 0.25 |
| ours | **93.99** | 0.28 |

(b)

| initialization | Acc. |
|---|---|
| random | 90.11 |
| constant | 93.91 |
| uniform | 93.87 |
| trunc_normal | 93.93 |
| normal (ours) | **93.99** |

(c)

| case | Acc. | Params. |
|---|---|---|
| w/o. scale | 93.49 | 0.18 |
| w/o. shift | 93.74 | 0.18 |
| only norm | 93.26 | 0.11 |
| scalar scale | 93.59 | 0.18 |
| ours | **93.99** | 0.28 |

(d)

Table 6: The impacts of different designs. (a) The impact of the number of layers with SSF-ADA. (b) The impacts of the different insertion locations of SSF-ADA. (c) The impacts of initialization. (d) The impacts of different components. Acc.: Top-1 accuracy (%); Params.: parameters (M).

**The impact of initialization.** We also investigate how different ways of initializing the scale and shift factors affect performance in Table 6c. In our experiments, we first randomly initialize both scale and shift parameters with a mean value of zero, but find that the performance is inferior (90.11%) and cannot converge in some experiments. After that, we randomly initialize the scale factor with a mean value of one and find better performance, which implies that the weights of a pre-trained model should not be completely disrupted in the fine-tuning, instead, we should start from this pre-trained model to optimize our model. Experiments show that using the normal initialization achieves the best performance, where the mean values of the scale factor and shift factor are one and zero, respectively.

**The impact of different components.** We also evaluate the impacts of different components in SSF-ADA and the results are shown in Table 6d. We find that removing the scale term yields worse performance than removing the shift term with the same trainable parameters, which shows that the scale term might be more important than the shift term. Also, note that the difference between 'w/o. scale' and the 'Bias' method in Table 5 is that we fine-tune the model with an additional shift term in 'w/o. scale', while 'Bias' fine-tunes the model based on the original biases, suggesting that fine-tuning the model in a res-like manner can obtain slightly better performance (93.49% *vs.* 93.39%). We also try to only fine-tune all scale and shift factors in the normalization layer (LN), or fine-tune the model with SSF but set the scale term as a scalar. These experiments yield inferior performance than SSF (93.26% *vs.* 93.99%, 93.59% *vs.* 93.99%), but could probably be considered as an alternative due to the fact that they only use about half of the trainable parameters of SSF.

### 4.4 Performance Comparisons on Robustness and OOD Datasets

We also conduct experiments to analyze the robustness and Out-Of-Distribution (OOD) ability of our SSF method with the following datasets: ImageNet-A, ImageNet-R and ImageNet-C. Please refer to Appendix for their details. We perform the robustness and OOD evaluation on these three datasets with the fine-tuned models on ImageNet-1K. All experimental results are listed in Table 7.

From this table, we can see that our SSF obtains better performance than VPT and other parameter-efficient fine-tuning methods on three datasets, which shows our fine-tuning method has stronger robustness and out-of-distribution generalization.

| Method \ Dataset | IN-1K (↑) | IN-A (↑) | IN-R (↑) | IN-C (↓) |
|---|---|---|---|---|
| Full fine-tuning | **83.58** | 34.49 | 51.29 | 46.47 |
| Linear probing | 82.04 | 33.91 | 52.87 | 46.91 |
| Adapter [21] | 82.72 | 42.21 | 54.13 | 42.65 |
| Bias [66] | 82.74 | 42.12 | 55.94 | 41.90 |
| VPT-Shallow [29] | 82.08 | 30.93 | 53.72 | 46.88 |
| VPT-Deep [29] | 82.45 | 39.10 | 53.54 | 43.10 |
| SSF (**ours**) | 83.10 | **45.88** | **56.77** | **41.47** |

Table 7: Performance comparisons on robustness and out-of-distribution datasets. 'IN' means ImageNet. The performance on IN-1K, IN-A and IN-R is evaluated in Top-1 accuracy (%). The performance on IN-C is evaluated in mCE (mean corruption error). The lower (↓), the better.

Furthermore, although SSF has lower accuracy than full fine-tuning on ImageNet-1K, the performance on ImageNet-A, ImageNet-R and ImageNet-C is better, which also shows the performance between ImageNet-1K and ImageNet-A/R/C is not absolutely positive relevant. Such improvements in robustness and OOD datasets might come from the fact that SSF freezes most of the pre-trained parameters, which maximally preserves the knowledge learned from the large-scale dataset and thus maintains a better generalization ability.

### 4.5 Visualization and Analysis

Although our goal is to modulate the features extracted by a pre-trained model, the scale and shift parameters are input-independent indeed. Therefore, these parameters can also be regarded as encoding information of the whole downstream dataset. After re-parameterization, these scale and

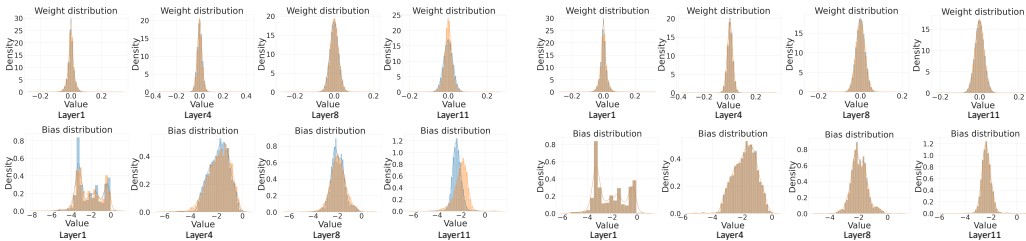

(a) Pre-trained model *vs*. Fine-tuned model via SSF.  (b) Pre-trained model *vs*. Full fine-tuned model.

Figure 4: Comparisons of parameter distribution between the original pre-trained model and different fine-tuning methods. The first row shows weight distribution and the second row is bias distribution. The blue histograms show the original pre-trained model, and the orange ones show the fine-tuned model via SSF in (a) and full fine-tuned model in (b).

shift parameters are absorbed into the original model weights. To better understand information learned by the SSF, we visualize the distributions of weights and biases before and after fine-tuning via SSF in Figure 4a. We can see that the scale and shift parameters adjust the original weights and biases, and change the distribution of weights and biases to fit the downstream task.

As a comparison, we also visualize the original weight distribution and the weight distribution after full fine-tuning in Figure 4b, from which we can find an interesting phenomenon that full fine-tuning does not change the distribution of weights and biases much, but probably only a small portion of the values is changed. It is worth noting that although SSF does not match the weight distribution of full fine-tuning, it achieves better performance (93.99% *vs*. 93.82% in Table 5) on CIFAR-100.

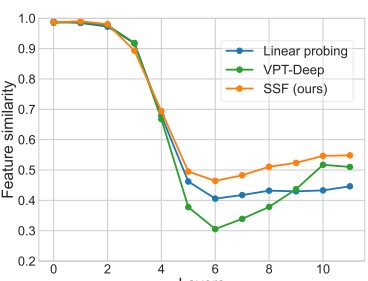

To further investigate why SSF can achieve superior performance, beyond weight distribution, we also visualize the feature similarities between full fine-tuning and linear probing, full fine-tuning and VPT-Deep, full fine-tuning and SSF, as shown in Figure 5. In the last layer, SSF has the most similar feature to full fine-tuning and the accuracy is also the closest. This shows that even if the weight distribution learned by SSF is different from full fine-tuning, SSF is also able to extract the features of the images in the downstream task very well, which validates the effectiveness of our method.

Figure 5: The visualization of the feature similarities between full fine-tuning and linear probing, full fine-tuning and VPT-Deep, full fine-tuning and SSF, in different layers of ViT-B/16.

## 5 Conclusion

In this paper, we focus on parameter-efficient fine-tuning and propose an SSF method to scale and shift the features extracted by a pre-trained model. The intuition behind our method comes from alleviating the distribution mismatch between upstream tasks and downstream tasks by modulating deep features. SSF surprisingly outperforms other parameter-efficient fine-tuning approaches with a small number of learnable parameters. Besides, the introduced scale and shift parameters during the fine-tuning can be merged into the original pre-trained model weights via re-parameterization in the inference phase, thereby avoiding extra parameters and FLOPs. With the proposed SSF method, our model obtains 2.46% (90.72% *vs*. 88.54%) and 11.48% (73.10% *vs*. 65.57%) performance improvement on FGVC and VTAB-1k in terms of Top-1 accuracy compared to the full fine-tuning but only fine-tuning about 0.3M parameters. Experiments on 26 image classification datasets in total and 3 robustness & out-of-distribution datasets with various model families (CNNs, Transformers, and MLPs) show the effectiveness of SSF, which establishes a new baseline.

## Acknowledgement

The authors acknowledge the support from the Singapore National Research Foundation ("CogniVision – Energy-autonomous always-on cognitive and attentive cameras for distributed real-time vision with milliwatt power consumption" grant NRF-CRP20-2017-0003) – `www.green-ic.org/CogniVision`. Xinchao Wang is the corresponding author.

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
