# A  Detailed Descriptions for the Evaluation Datasets

## A.1  Image Classification

We show the detailed descriptions of image classification as follows. The train/val/test splits and the classes are shown in Table 1.

| Dataset | Description | #Classes | Train size | Val size | Test size |
|---|---|---|---|---|---|
| *Fine-Grained Visual Classification (FGVC)* | | | | | |
| CUB-200-2011 [31] | Fine-grained bird species recognition | 200 | 5,394* | 600* | 5,794 |
| NABirds [29] | Fine-grained bird species recognition | 55 | 21,536* | 2,393* | 24,633 |
| Oxford Flowers [27] | Fine-grained flower species recognition | 102 | 1,020 | 1,020 | 6,149 |
| Stanford Dogs [20] | Fine-grained dog species recognition | 120 | 10,800* | 1,200* | 8,580 |
| Stanford Cars [8] | Fine-grained car classification | 196 | 7,329* | 815* | 8,041 |
| *Visual Task Adaptation Benchmark (VTAB-1k) [34]* | | | | | |
| CIFAR-100 [21] | | 100 | | | 10,000 |
| Caltech101 [7] | | 102 | | | 6,084 |
| DTD [4] | | 47 | | | 1,880 |
| Flowers102 [27] | Natural | 102 | 800/1,000 | 200 | 6,149 |
| Pets [28] | | 37 | | | 3,669 |
| SVHN [26] | | 10 | | | 26,032 |
| Sun397 [32] | | 397 | | | 21,750 |
| Patch Camelyon [30] | | 2 | | | 32,768 |
| EuroSAT [14] | | 10 | | | 5,400 |
| Resisc45 [3] | Specialized | 45 | 800/1,000 | 200 | 6,300 |
| Retinopathy [12] | | 5 | | | 42,670 |
| Clevr/count [19] | | 8 | | | 15,000 |
| Clevr/distance [19] | | 6 | | | 15,000 |
| DMLab [1] | | 6 | | | 22,735 |
| KITTI/distance [9] | | 4 | | | 711 |
| dSprites/location [25] | Structured | 16 | 800/1,000 | 200 | 73,728 |
| dSprites/orientation [25] | | 16 | | | 73,728 |
| SmallNORB/azimuth [22] | | 18 | | | 12,150 |
| SmallNORB/elevation [22] | | 9 | | | 12,150 |
| *General Image Classification Datasets* | | | | | |
| CIFAR-100 [21] | General image classification | 100 | 50,000 | - | 10,000 |
| ImageNet-1K [6] | | 1,000 | 1,281,167 | 50,000 | 150,000 |
| *Robustness and Out-of-Distribution Dataset* | | | | | |
| ImageNet-A [17] | | 200 | | 7,500 | |
| ImageNet-R [15] | Robustness & OOD | 200 | | 30,000 | |
| ImageNet-C [16] | | 1,000 | | $75 \times 50,000$ | |

Table 1: The statistics of the various datasets. *: Since there are no public train/val splits in these datasets, we follow VPT [18] for random train/val split. This table is partially borrowed from VPT [18].

*FGVC*. Following VPT [18], we employ five Fine-Grained Visual Classification (FGVC) datasets to evaluate the effectiveness of our proposed SSF, which consists of CUB-200-2011 [31], NABirds [29], Oxford Flowers [27], Stanford Dogs [20] and Stanford Cars [8].

*VTAB-1k*. VTAB-1k benchmark is introduced in [34], which contains 19 tasks from diverse domains: i) Natural images that are captured by standard cameras; ii) Specialized images that are captured by non-standard cameras, *e.g.*, remote sensing and medical cameras; iii) Structured images that are synthesized from simulated environments. This benchmark contains a variety of tasks (*e.g.*, object counting, depth estimation) from different image domains and each task only contains 1,000 training samples, thus is extremely challenging.

*General Image Classification Datasets*. We also validate the effectiveness of SSF on general image classification tasks. We choose the CIFAR-100 [21] and ImageNet-1K [6] datasets as evaluation datasets, where CIFAR-100 contains 60,000 images with 100 categories. ImageNet-1K contains 1.28M training images and 50K validation images with 1,000 categories, which are very large datasets for object recognition.

## A.2 Robustness and OOD

*ImageNet-A* is introduced in [17], where 200 classes from 1,000 classes of ImageNet-1K are chosen and the real-world adversarial samples that make the ResNet model mis-classified are collected.

*ImageNet-R* [15] contains rendition of 200 ImageNet-1K classes and 30,000 images in total.

*ImageNet-C* [16] consists of the corrupted images, including noise, blur, weather, *etc.* The performance of model on ImageNet-C show the robustness of model.

## A.3 Detection and Segmentation

We also conduct experiments on downstream tasks beyond image classification, such as object detection, instance segmentation and semantic segmentation. We employ the COCO dataset [23] for evaluation based on mmdetection [2] framework for the object detection and instance segmentation. COCO contains 118K training images for training and 5K images for validation, which is one of the most challenging object detection datasets. We use Mask R-CNN [13] with Swin Transformer backbone to perform our experiments, following the same training strategies as Swin Transformers [24]. For semantic segmentation, we employ the ADE20K dataset [35] for evaluation based on mmsegmentation [5] framework. ADE20K contains 20,210 training images and 2,000 validation images. Following Swin Transformer [24], we use UperNet [33] with Swin Transformer backbone . All models are initialized with weights pre-trained on ImageNet-1K for detection and segmentation.

| Dataset / Method | COCO with Mask R-CNN | | | | | | ADE20K with UPerNet | |
|---|---|---|---|---|---|---|---|---|
| | $AP^b$ | $AP^b_{50}$ | $AP^b_{75}$ | $AP^m$ | $AP^m_{50}$ | $AP^m_{75}$ | mIoU | MS mIoU |
| Full fine-tuning | 43.7 | 66.6 | 47.7 | 39.8 | 63.3 | 42.7 | 44.5 | 45.8 |
| Linear probing | 31.7 | 55.7 | 32.5 | 31.2 | 53.0 | 32.2 | 35.7 | 36.8 |
| VPT-Deep [18] | 33.8 | 57.6 | 35.3 | 32.5 | 54.5 | 33.9 | 37.0 | 37.9 |
| SSF (ours) | 34.9 | 58.9 | 36.1 | 33.5 | 55.8 | 34.7 | 38.9 | 39.8 |

Table 2: Performance of different fine-tuning methods on the COCO val2017 dataset and ADE20K dataset, where $AP^b$ and $AP^m$ are the average precision of object detection and instance segmentation, respectively. mIoU and MS mIoU show single-scale and multi-scale inference of semantic segmentation.

# B Experiments on Detection and Segmentation

We also conduct experiments on broader downstream tasks, *e.g.*, object detection, instance segmentation, and semantic segmentation. For object detection and instance segmentation, we perform experiments on the COCO dataset with Mask R-CNN [13], where Swin-T pre-trained on ImageNet-1K is adopted as the backbone. The specific hyper-parameter setup and data augmentation refer to Swin Transformer [24] and mmdetection [2]. We perform i) full fine-tuning; ii) linear probing, where the weights at the backbone layers are frozen and only weights at the neck and head layers are updated; iii) VPT-Deep; iv) SSF. All models are trained with 1x schedule (12 epochs). The results are shown in Table 2. We can see that SSF outperforms linear probing and VPT-Deep [18] on the COCO dataset in terms of object detection and instance segmentation. For semantic segmentation, we perform experiments on the ADE20K dataset with UperNet [33] and Swin-T pre-trained on ImageNet-1K. The results in Table 2 show that SSF outperforms linear probing and VPT-Deep [18]. However, for both datasets, SSF still has a large gap compared to the full fine-tuning, which might be due to the fact that detection and segmentation tasks are fundamentally different from classification tasks. Only fine-tuning a few parameters in the backbone will result in inferior performance. How to introduce trainable parameters for parameter-efficient fine-tuning in object detection and segmentation will be the future work.

# C Visualizations

## C.1 Feature Distribution

We also visualize the feature distribution of different fine-tuning methods via t-SNE on the CIFAR-100 dataset. All fine-tuning methods are based on a ViT-B/16 pre-trained on the ImageNet-21K datasets. The results are shown in Figure 1. Our SSF achieves better feature clustering results compared to linear probing and VPT-Deep. Besides, since our method and full fine-tuning have similar accuracy

(93.99% *vs.* 93.82%), it is difficult to distinguish them in terms of feature distribution, which also shows the effectiveness of our method.

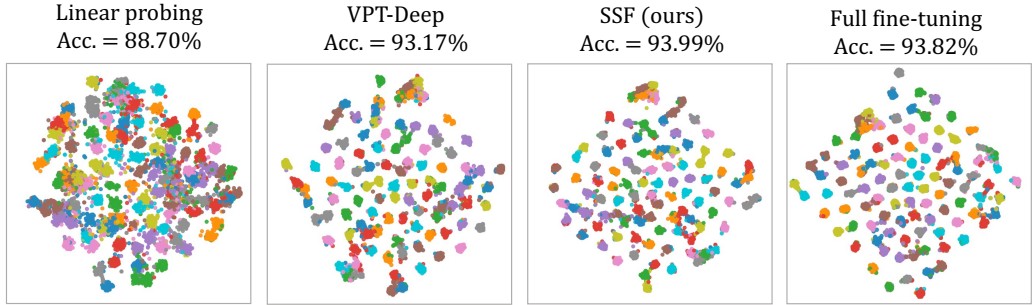

Figure 1: t-SNE visualization of different fine-tuning methods, including linear probing, VPT-Deep, our SSF, and full fine-tuning (best viewed in color).

## C.2 Attention Map

We also visualize the attention maps of different fine-tuning methods, as shown in Figure 2. All models are fine-tuned on ImageNet-1K with ViT-B/16 pre-trained on ImageNet-21K. The specific experimental results refer to the main text. We find that VPT-Deep has more concentrated attention on the object in some images (*e.g.*, the first two lines), but lacks suitable attention on some other images (*e.g.*, the last two lines). In contrast, SSF tends to obtain attention similar to the full fine-tuning but also generates the failure prediction such as the second row.

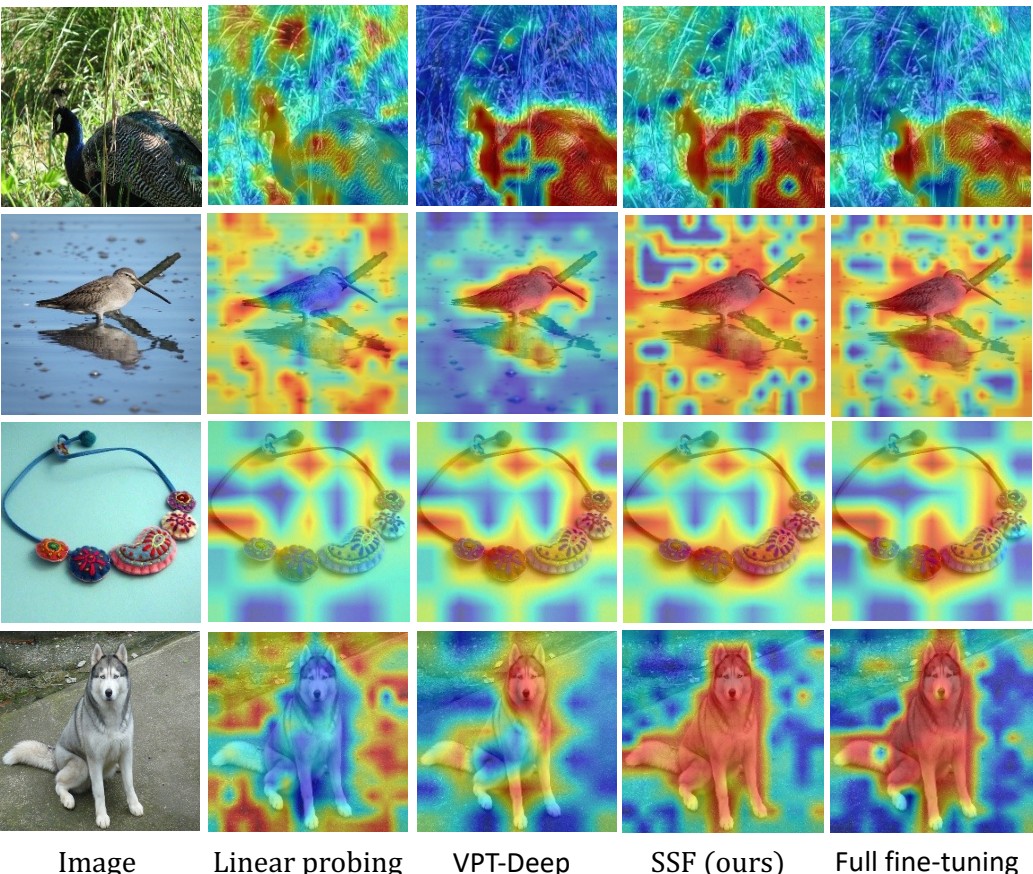

Figure 2: Visualization of attention maps. From left to right, each column shows the RGB image, linear probing, VPT-Deep, our SSF and full fine-tuning.

# D   Limitations and Societal Impacts

Regarding the limitations of this work, we currently focus on sharing backbone parameters among different tasks while treating each task independently of the rest of the tasks involved. However, some recent papers (*e.g.*, [11, 10]) show that by correlating multiple tasks together during the fine-tuning, the performance for every single task can be further improved. However, recent works treat this relationship among tasks as a black box that inevitably suffers a huge computational cost. Thus, we believe an efficient method to find positive task relationships could be a meaningful direction for further exploration.

This work has the following societal impact. SSF can effectively save parameters compared to the full fine-tuning so that the approach can quickly transfer large models pre-trained on large datasets to downstream tasks, which saves computational resources and carbon emissions. Thanks to the linear transformation and re-parameterization, we do not need to change the deployed backbone architecture when the model is transferred to the downstream task. Only a set of weights need to be replaced, which is also more convenient compared to the methods that introduce additional parameters such as VPT [18]. However, like other fine-tuning methods, SSF is also based on a pre-trained model, which will probably also cause a violation of the use of fine-tuning methods if this upstream pre-trained model is trained on some illegal data.