# OpenReview forum: "Scaling & Shifting Your Features: A New Baseline for Efficient Model Tuning"
_NeurIPS.cc/2022/Conference — NeurIPS 2022 Accept_

### Official Review · Reviewer_eK8a · 2022-07-11

**Rating:** 7
**Confidence:** 5
**Soundness:** 4 excellent
**Presentation:** 3 good
**Contribution:** 4 excellent

**Summary:**

The paper introduces a parameter-efficient transfer learning method for vision transformer, named LInear Feature Scalability (LIFTs). It inserts a linear-structured adapter layer to the transformer block, while keeping other model parameters unaltered. The micro and macro insertion positions are carefully searched through experiments. The trained linear adapter layer can be fused into the original network structure by applying the reparameterization tricks, without additional computational overhead at inference time. LIFT shows promising results on a variety of evaluations.

**Questions:**

1.	As the paper put a lot of effort on the where to insert the LIFT block, the current version seems to a simplified Neural Architecture Search with grid search strategy. For example, 3 macro designs, called LIFTs-Shallow, LIFTs-Shared and LIFTs-Deep are evaluated. But there are still other hybrid micro and macro designs that are untested. I am wondering whether the author tried some NAS techniques for such combinational optimization problem.

2.	In table 3, why not include the ViT-B structure for CIFAR-100 and ImageNet1k experiments? Since the Visual prompt tuning was originally designed for ViT-structured network, I believe it is more appropriate and important to report the results for CIFAR and ImageNet with ViT, since it provides a fair test bed for two methods.

**Limitations:**

Please provide some discussion on the limitations and societal impact in the paper.

**Strengths And Weaknesses:**

Strengths:
1.	Simple but efficient design. The linear-structured adapter can be easily implemented and plugged into any existing vision transformer structure without hustle.
2.	The proposed LIFT adapter takes the idea of reparameterization method, thus avoid the additional inference cost.
3.	The performance improvement is encouraging.

Weakness:
1.	Since the paper propose a new adapter-based architecture for transfer learning, please compare the computational complexity (Table 1) and model performance (Table 3 and Table 4) with original adapter-tuning and recent vision adapter structures. I do not see the results in the paper.

---

> ### Author Response · Authors · 2022-08-02
> **Response to Reviewer eK8a (1/2)**
>
> Thanks for your valuable suggestions.
>
> **Q1: Since the paper propose a new adapter-based architecture for transfer learning, please compare the computational complexity (Table 1) and model performance (Table 3 and Table 4) with original adapter-tuning and recent vision adapter structures. I do not see the results in the paper.**
>
> A1: Thanks for your suggestions. Following VPT [1], we have added the complexity analysis of the Adapter [2] in Table 1 of the revised version. We also add experiments to show the performance of Adapter [2, 4] and Bias [3] with different architectures (ViT-B/16, Swin-B, ConvNext-B) in Table 3 of the revised version, where our LIFTs still achieves superior results compared to other baselines. The running time and memory are shown in Table 4.
>
> **Q2: As the paper put a lot of effort on the where to insert the LIFT block, the current version seems to a simplified Neural Architecture Search with grid search strategy. I am wondering whether the author tried some NAS techniques for such combinational optimization problem.**
>
> A2: This is a good question and we also agree that incorporating NAS algorithm is a reasonable extension. Currently, we are using a method similar to grid search to find insertion locations, which is a bit like NAS. However, NAS requires an additional search phase, and the structure searched in one task is not necessarily applicable to another task [5]. Some recent works [6, 7] indeed show that, given a task, a multitask system can automatically learn the best path for the specific task, thereby activating only a specific module for this task in a large system, similar to pathway [8]. However, this could be extremely computational expensive and considerable amount of extra works. Therefore, we leave it as a future and separate work of building a larger system and then finding task-specific activation paths via NAS, to keep the current world focus on the problem of efficient fine tuning.
>
>
> Back to the current approach, we also only explore the structure on CIFAR-100, which might not be the optimal structure on other datasets (Even if we use NAS, we cannot also guarantee that the structure searched on one dataset will be optimal on other datasets), but we are more interested in whether our approach works well and obtains superior performance. Considering your suggestion, we conduct an experiment where the lifts-ada is inserted into all layers and all positions. Such an approach is an upper bound of performance. We conduct experiments on CIFAR-100 and ImageNet-1K with pre-trained ViT-B/16 and show that such an approach achieves higher performance than lifts-deep (CIFAR-100: 93.24 vs. 92.76, ImageNet-1K: 83.10 vs. 82.82). Despite lifts-ada is inserted into all layers and all positions, the inference phase still does not require any additional parameters due to the re-parameterization.
>
> **Q3: It is more appropriate and important to report the results for CIFAR and ImageNet with ViT in Table 3.**
>
> A3: Thanks for your suggestions. We have added the results for CIFAR and ImageNet in Table 3 with pre-trained ViT-B/16 following VPT [1]. The specific results are as follows.
>
> |                              |CIFAR-100     |  ImageNet-1K |
> | :--------------------: |:---------------: | :------------: |
> |   Full fine-tuning   |         93.69       |        83.58   |
> | Linear probing      |          87.28     |          80.31 |
> |Adapter [2] 		|    92.42           |    82.66   |
> |Bias [3]                	|    92.21 	        |82.75      |
> |VPT-Shallow [1] 	|    91.28 	        |81.43     |
> |VPT-Deep [1] 		|    92.01           |    82.69  |
> |LIFTs (ours) 		|    92.76	         |82.82    |
>
>
> From this table, we can see that, although there is a gap between LIFTs and full fine-tuning, our method still outperforms other parameter-efficient fine-tuning methods.

---

> ### Author Response · Authors · 2022-08-02
> **Response to Reviewer eK8a (2/2)**
>
> **Q4: Please provide some discussion on the limitations and societal impact in the paper.**
>
> A4: Regarding the limitations of this work, we currently focus on sharing backbone parameters among different tasks while treating each task independent of the rest of the tasks involved. However, some recent papers (e.g., [1, 2]) show that by correlating multiple tasks together during fine-tuning, the performance for each single task can be further improved. However, recent works treat this relationship among tasks as a black box in which investable suffers a huge computational cost. Thus, we believe an efficient method to find positive task relationships could be a meaningful direction for further exploration.
>
> This work has the following societal impact. LIFTs can effectively save parameters and training time compared to full fine-tuning, so that the approach can quickly transfer large models pre-trained on large datasets to downstream tasks, which saves computational resources and carbon emissions. Thanks to the linear transformation and re-parameterization, there is also no need to change the deployed model architecture when the model is transferred to the downstream task. Only a set of weights need to be replaced, which is also more convenient compared to the methods that introduce additional parameters such as VPT [1]. However, like other fine-tuning methods, LIFTs is also based on a pre-trained model, which will probably also cause a violation of the use of fine-tuning methods if this upstream pre-trained model is trained on some illegal data.
>
> [1] ﻿ Menglin Jia, Luming Tang, Bor-Chun Chen, Claire Cardie, Serge Belongie, Bharath Hariharan, Ser-Nam Lim. Visual Prompt Tuning. ECCV2022.
>
> [2] Neil Houlsby, Andrei Giurgiu, Stanislaw Jastrzebski, Bruna Morrone, Quentin de Laroussilhe, Andrea Gesmundo, Mona Attariyan, Sylvain Gelly. Parameter-Efficient Transfer Learning for NLP. ICML2019.
>
> [3] Elad Ben Zaken, Yoav Goldberg, Shauli Ravfogel. BitFit: Simple Parameter-efficient Fine-tuning for Transformer-based Masked Language-model. ACL2022.
>
> [4] Shoufa Chen, Chongjian Ge, Zhan Tong, Jiangliu Wang, Yibing Song, Jue Wang, Ping Luo.AdaptFormer: Adapting Vision Transformers for Scalable Visual Recognition, arXiv preprint, 2022.
>
> [5] Han Cai, Ligeng Zhu, Song Han.ProxylessNAS: Direct Neural Architecture Search on Target Task and Hardware. ICLR2019.
>
> [6] ﻿Andrea Gesmundo, Jeff Dean. ﻿muNet: Evolving Pretrained Deep Neural Networks into Scalable Auto-tuning Multitask Systems, arXiv preprint, 2022.
>
> [7] ﻿Andrea Gesmundo, ﻿Jeff Dean. ﻿An Evolutionary Approach to Dynamic Introduction of Tasks in Large-scale Multitask Learning Systems, arXiv preprint, 2022.
>
> [8] Paul Barham et al. Pathways: Asynchronous Distributed Dataflow for ML. MLSys2022.

---

### Official Review · Reviewer_r389 · 2022-07-11

**Rating:** 7
**Confidence:** 5
**Soundness:** 3 good
**Presentation:** 3 good
**Contribution:** 3 good

**Summary:**

A new fine-tuning method for vision transformer is proposed in this manuscript. The authors have explored the Linear Feature Scalability of vision transformers, which defined as the fine-tuning performance with only linear transformation of the feature map. A simple module named LIFTs-ADA is designed for a fine-tuning, where some additional linear transformation operations are injected after the MLP and MSA. Re-parameterization is designed to make the fine-tuning parameter-free during inference. It has shown its efficiency on different tasks and models.


**Questions:**

Please refer to the weaknesses part

**Limitations:**

The limitations are mainly about the unclear statement of some parts and the insufficient analysis about the learning behaviors (the different visualizations of the linear transformation).


**Strengths And Weaknesses:**

Strengths:
1, The definition of feature scalability and the observation that applying only a linear transformation of the feature map can lead to great performances on different downstream tasks are interesting.
2, The proposed method is quite simple yet efficient for vision transformers with respect to the naive fine-tuning and vision prompt fine-tuning. The additional FLOPS of the proposed method is indeed lower than that of the prompt-based method.
3, The Re-parameterization trick can neurally integrate the linear transformation learned for different tasks into the original linear layer of transformer, leading to a zero-overhead inference.
4, The ablation studies about different configuration of ADA is good.

Weaknesses:
1, Some parts of the manuscript is unclear. For example, line 195, A_2^2 should be A_i^2. It is unclear whether the transformation A is different among different attention heads within the same layer.
2, For the Fig. 3 (b), it is recommended to provide details explanation about how the similarity matrix is computed. Also, it would be interesting to show the dissimilarity matrix of the linear transformation before and after fine-tuning, since they are comparable after re-parameterization.
3, For Tab. 2, why the Tuned/Total rates are so different between NABirds dataset and others?

---

> ### Author Response · Authors · 2022-08-02
> **Response to Reviewer r389**
>
> Thanks for your valuable comments.
>
> **Q1: Some parts of the manuscript is unclear. For example, line 195, A_2^2 should be A_i^2. It is unclear whether the transformation A is different among different attention heads within the same layer.**
>
> A1: Thanks for your suggestion. We have refined this description as Line 194 of the revised version. As you mentioned, A_2^2 should be A_i^2, and the transformation A is different among different attention heads.
>
>
> **Q2: For the Fig. 3 (b), it is recommended to provide details explanation about how the similarity matrix is computed. Also, it would be interesting to show the dissimilarity matrix of the linear transformation before and after fine-tuning, since they are comparable after re-parameterization.**
>
> A2: Thanks for the good question. For Fig.3 (b), we visualize the similarity matrix of LIFTs-ADA in different layers of LIFTs-Deep. In each layer, there exist a vector of scale and a shift variables of LIFTs-ADA, as shown in Alg. 1 of the submitted version. We compute the similarity of the scale factor across different layers to show their weak correlation. That is to say, they learn independent representation in each layer rather than interdependent and redundant features.
>
> We found the proposed dissimilarity quite interesting. However, the model weights parameters are extremely sensitive. To verify this, we conduct a simple ablation experiment according to your suggestion as shown in the following parts. First, we choose three set of fine-tuned model weights for comparison: full fine-tuning, linear probing, and the proposed LIFTs. After getting the weights, we compute the dissimilarity of weights between full fine-tuning and linear probing. After that, we compute the dissimilarity of weights between full fine-tuning and LIFTs. Finally, we compare both dissimilarity values and find LIFTs has a higher value than linear probing, but LIFTs has higher accuracy. It means that, even though the accuracies of the two networks are similar, the weights might be very different. We have also conducted another step of experiments for a sanity check where we train the same model with different seeds. Even with similar final accuracy, the learned model weights can be quite different.
>
> Additionally, inspired by your comment, we find the feature similarity might better depict the similarity of the model [1]. It is possible that even though the weights of the two models are very different, the extracted features are similar. Therefore, we show the curve of CKA w.r.t. the layers in Fig.3 (b). We compute the feature similarity between full fine-tuning and LIFTs, full fine-tuning and linear probing, and full fine-tuning and VPT-Deep separately. Near the classification layer, LIFTs have higher feature similarity to full fine-tuning compared with linear probing and VPT-Deep, which shows the effectiveness of our method.
>
>
> **Q3: For Tab. 2, why the Tuned/Total rates are so different between NABirds dataset and others?**
>
> A3: In Table 2, we compute all tuned parameters, which also contain the head of the network and it’s number of parameters is proportional to the number of classes. NABirds only has 55 classes but other datasets have 200 (CUB), 102 (Flowers), 120 (Stanford Dogs), and 196 (Stanford Cars) classes, respectively. NABirds has fewer head parameters than other datasets but the total parameters are similar because the same ViT-B/16 backbone is employed. Thus, the tuned/total rate of NABirds is much less than other datasets.
>
> [1] Simon Kornblith, Mohammad Norouzi, Honglak Lee, Geoffrey Hinton. Similarity of Neural Network Representations Revisited. ICML2019.

---

### Official Review · Reviewer_RByJ · 2022-07-12

**Rating:** 7
**Confidence:** 3
**Soundness:** 3 good
**Presentation:** 4 excellent
**Contribution:** 4 excellent

**Summary:**

In this submission, instead of optimizing all the parameters to adapt to a new task, the authors propose a LInear Feature Scalability (LIFTs) method that keeps all the model parameters from the pre-trained network but only learns the task-specific linear feature scalability adaption layer for each block. In this way, the authors alleviate the dilemma of useful information elimination and expensive computation burden as in previous all-parameter-optimized fine-tuning approaches, making the model readily and effectively reusable across various tasks. Experiments on both the fine-grained visual classification and general classification datasets convincingly demonstrate the superiority of the proposed LIFTs over prior works. Further results on segmentation and detection are also provided in the supplement.

**Questions:**

- Could you analyze the limitations of the proposed method? It will be helpful for follow-up research on this line of fine-tuning schemes with the feature scalability.

**Limitations:**

Yes, limitations have been discussed. I do not find the potential negative societal impact of this work.

**Strengths And Weaknesses:**

[Strengths]
- The proposed module is plug-and-play, which is easy to be incorporated in various architectures. The motivation is clear. The authors analyze the flaws of the existing full-parameter fine-tuning methods in details, and then demonstrate their solution of LIFTs accordingly.

- The detailed algorithm workflow with the key code is provided, which makes it easier to understand the method details. Detailed discussions on the complexity are given in the method section, which is critical to understanding the contributions of the proposed LIFTs. The paper is easy to follow. The illustrations are quite clear, by firstly giving a definition of the key term of the feature scalability in this paper.

- Extensive ablation studies and visualization results are provided, by exploring the impacts of various designs.

[Weaknesses]
- Please explicitly denote in Tab. 4 on what device the running time is obtained.

- The authors are suggested to use the same font across all the figures.

- It will be great if the authors could discuss the limitations of the proposed method, which may showcase the future directions on this line of research to some extent.

- Grammars: Ln 329 “ … which makes the network applied to downstream tasks not require any additional parameters and FLOPs …”

---

> ### Author Response · Authors · 2022-08-02
> **Response to Reviewer RByJ**
>
> Thanks for your valuable comments and recognization.
>
> **Q1: Please explicitly denote in Tab. 4 on what device the running time is obtained.**
>
> A1: Thanks for your suggestion. For the training stage, we employ a batch size of 16 and mixed precision training. For the inference stage, the batch size is 1. The running time in Table 4 is measured in a single GeForce RTX 2080Ti GPU with 11G memory in total. See Line 292-296 of the revised version.
>
>
> **Q2: It will be great if the authors could discuss the limitations of the proposed method, which may showcase the future directions on this line of research to some extent.**
>
> A2: Thanks for your suggestions. Currently, we focus on sharing backbone parameters among different tasks while treating each task independent of the rest of the tasks involved. However, some recent papers (e.g., [1, 2]) show that by correlating multiple tasks together during fine-tuning, the performance for each single task can be further improved. However, recent works treat this relationship among tasks as a black box in which investable suffers a huge computational cost. Thus, we believe an efficient method to find positive task relationships could be a meaningful direction for further exploration.
>
> [1] ﻿Andrea Gesmundo, Jeff Dean. ﻿muNet: Evolving Pretrained Deep Neural Networks into Scalable Auto-tuning Multitask Systems, arXiv preprint, 2022.
>
> [2] ﻿Andrea Gesmundo, ﻿Jeff Dean. ﻿An Evolutionary Approach to Dynamic Introduction of Tasks in Large-scale Multitask Learning Systems, arXiv preprint, 2022.

---

### Official Review · Reviewer_SF4A · 2022-07-15

**Rating:** 7
**Confidence:** 4
**Soundness:** 3 good
**Presentation:** 2 fair
**Contribution:** 3 good

**Summary:**

The authors propose a simple linear adapter for parameter-efficient fine-tuning. Unlike previous methods that involve complex non-linear operations or prompting, they explore linear feature scalability of vision transformers and propose LIFTs-ADA. Furthermore, their linear adapters can be merged or subsumed into the original modules of the network during inference stage making them not requiring any additional parameters and FLOPs. Their experiments show that their proposed method performs at par or better than previous competing methods such as VPT with substantially less number of parameters during fine-tuning for downstream tasks.




**Questions:**

1. Please review and explain this section better (lines 175-177)

Here, we consider that g is a linear function for its simplicity of the linear function but the lack of comprehensive investigation and the properties of the linear function make it possible to be merged with other operations of the network in the inference phase.

I do not understand this line. Looks like 2 unrelated sentences have been merged.

2. The claim (lines 27-27) that the you do not have to store the pretrained seperately for each fine-tuned model seems flawed. If the Linear adapters are subsumed once into the pre-trained network during inference stage, they you would have to do this for each downstream task and store a different model for each downstream task.

**Limitations:**

1. Lack of results on robustness or OOD performance.
2. Writing quality at some places is not up-to the standards.
3. Lack of theoretical grounding (but this can be excused).
4. For each downstream task, you still have to store a separate model since the parameters of the linear adapters are subsumed into original parameters of the pre-trained network thus updating the parameters effectively.

**Strengths And Weaknesses:**

Strengths

1. Method is simple, effective and easy to use.
2. Inference stage does not require any additional parameters and FLOPs, thereby reducing computational costs. They also provide numbers (#params and #FLOPs) for comparison with another parameter-efficient fine-tuning method such as VPT.
3. Authors have performed a comprehensive set of experiments with various datasets and downstream tasks to support their method and claims.
4. Authors also perform a series of ablations to understand the effectiveness of locations for placing their linear adapters.

Weaknesses
1. Paper is poorly written in some places. Some specific lines 103, 175, 176, 177. In general the paper lacks the written maturity of a Neurips paper.
2. Lack of theoretical grounding. But their simplicity of idea and empirical effectiveness seem to be sufficient according to me.
3. Lack of study on robustness and OOD performance.

---

> ### Author Response · Authors · 2022-08-02
> **Response to Reviewer SF4A (1/2)**
>
> We truly appreciate the reviewer for the constructive comments.
>
> **Q1: Lack of theoretical grounding, but their simplicity of the idea and empirical effectiveness seems to be sufficient according to me.**
>
> A1: Thanks for the comment. The intuition behind our idea is feature distribution alignment [1] which is under the theoretical framework of feature distribution matching. Specifically, the parameters tuned per block are to align the first-order (mean) and second-order (variance) statistics of the feature distribution to the target data. We design extensive experiments to empirically show the effectiveness of our method, which, hopefully, will establish a solid baseline of parameter-efficient fine-tuning. It is indeed hard to theoretically prove the behaviour of a deep neural network model. In our future work, as pointed out by you, we will attempt to look into the theoretical groundings.
>
>
> **Q2: Lack of study on robustness and OOD performance.**
>
> A2: Thanks for the suggestions. Here we conduct experiments to evaluate the robustness and OOD performance. The results are shown in Table 5 of the revised version. We directly perform the evaluation on the ImageNet-A,  ImageNet-R and ImageNet-C datasets with the fine-tuned models on ImageNet-1K. ImageNet-A and ImageNet-R are measured by Acc@1. The performance on ImageNet-A and ImageNet-R shows the OOD prediction of models. ImageNet-C is measured by mCE ($\downarrow$). The performance on ImageNet-C shows the robustness of the models. For your convenience, we also list this table as follows.
>
> |   		     | ImageNet-1K    | ImageNet-A    | ImageNet-R    | ImageNet-C  |
> | :-----------------:  | :-----------------: | :---------------: | :----------------: | :--------------: |
> | Full fine-tuning     |     83.58      |          34.49       |        51.29          |       46.47         |
> | Linear probing      |     80.31 	     |	29.43	       |	50.83	           |	       49.04	|
> | VPT-Deep             |     82.69       |	44.03	      |	53.41              |	     42.05	|
> | LIFTs (ours)          |     82.82       |         45.88	|	56.77  |      41.47	|
>
>
>
> We have two findings from this table: i) our LIFTs obtains better performance than VPT on both datasets, which shows our fine-tuning method has stronger robustness and out-of-distribution generalization; ii) although LIFTs has lower accuracy than full fine-tuning on ImageNet-1K, the performance on ImageNet-A, ImageNet-R and ImageNet-C are better. As pointed out in paper [2, 3, 4], the performance between ImageNet-1K and ImageNet-A (ImageNet-C) is not absolutely relevant. We believe such improvements in robustness and OOD datasets might come from the fact that LIFTs freeze most of the pre-trained parameters and thus maximally preserve the knowledge learned on the large dataset for pre-training and thus maintain a better generalization capability. We found this is an extremely interesting point and will explore more on this in a separate work.
>
>
> **Q3: Please review and explain this section better (lines 175-177).**
> Here, we consider that g is a linear function for its simplicity of the linear function but the lack of comprehensive investigation and the properties of the linear function make it possible to be merged with other operations of the network in the inference phase.
>
> A3: Thanks for the suggestions. We have added more explanations regarding this part. Please refer to Line 174-176 in the revised version. The refined description is  ‘Here, we consider that g is a linear function for the simplicity of the linear function, and the properties of the linear function make it possible to be merged with other operations of the network in the inference stage.’. Such an idea is inspired by [5, 6]. One of the representative techniques is batch normalization folding used in the model compression algorithms. The parameters introduced by the batch normalization layers are fused into the convolution layers which are usually implemented before them. We deploy a similar strategy such that during the inference phase, the parameters introduced during the training phase are merged into the linear layer defined in the baseline model. In this manner, LIFTs achieves state-of-the-art performance without extra parameters and computation overhead during the inference stage.

---

> ### Author Response · Authors · 2022-08-02
> **Response to Reviewer SF4A (2/2)**
>
> **Q4: The claim (lines 27-27) that you do not have to store the pretrained separately for each fine-tuned model seems flawed.**
>
> A4: Thanks for the valuable suggestions. We have revised this part to make it clearer. Please refer to Line 26-27 of our revised version. Specifically, We divide the storage of model parameters into two different phases: the undeployed phase and the deployed phase. In the undeployed phase, compared to the full fine-tuning where all model parameters need to be stored for each task, our LIFTs only need to store a few parameters. When the model is downloaded from the server and needs to be sped up for a single target task, which is termed the deployment phase, the small number of parameters can be absorbed into the large model via re-parameterization while not changing the model architecture as VPT [7] does.
>
> [1] Baochen Sun, Jiashi Feng, Kate Saenko. Return of Frustratingly Easy Domain Adaptation. AAAI2016.
>
> [2] Dan Hendrycks, Norman Mu, Ekin D. Cubuk, Barret Zoph, Justin Gilmer, Balaji Lakshminarayanan. AugMix: A Simple Data Processing Method to Improve Robustness and Uncertainty. ICLR2020.
>
> [3] Enze Xie, Wenhai Wang, Zhiding Yu, Anima Anandkumar, Jose M. Alvarez, Ping Luo. SegFormer: Simple and Efficient Design for Semantic Segmentation with Transformers. NeurIPS2021.
>
> [4] Daquan Zhou, Zhiding Yu, Enze Xie, Chaowei Xiao, Anima Anandkumar, Jiashi Feng, Jose M Alvarez. Understanding The Robustness in Vision Transformers. ICML2022.
>
> [5] Benoit Jacob, Skirmantas Kligys, Bo Chen, Menglong Zhu, Matthew Tang, Andrew Howard, Hartwig Adam, and Dmitry Kalenichenko. Quantization and training of neural networks for efficient integer-arithmetic-only inference. CVPR2018.
>
> [6] Xiaohan Ding, Xiangyu Zhang, Ningning Ma, Jungong Han, Guiguang Ding, and Jian Sun. Repvgg: Making vgg-style convnets great again. CVPR2021.
>
> [7] ﻿ Menglin Jia, Luming Tang, Bor-Chun Chen, Claire Cardie, Serge Belongie, Bharath Hariharan, Ser-Nam Lim. Visual Prompt Tuning. ECCV2022.

---

> ### Comment · Reviewer_SF4A · 2022-08-09
> **Rebuttal Acknowledgement**
>
> Thank you authors for the detailed responses to my questions. I have reviewed them and they seem to answers my concerns. I have updated my rating accordingly.

---

> > ### Author Response · Authors · 2022-08-09
> > **Response to Reviewer SF4A**
> >
> > Thanks for your valuable comments and suggestions! We sincerely appreciate your recognition and constructive comments to improve our work.

---

### Author Response · Authors · 2022-08-02
**Response to AC and all reviewers**

Dear AC and all reviewers,

We are grateful to AC for organizing the review of our paper and appreciate all the reviewers for the valuable comments and recognition of our work. We have revised our manuscript carefully and improved the proofreading based on all reviewers’ comments. The revised version has been updated online. The updated sections are marked in the magenta font. The responses to each reviewer are as follows. We will also continue to revise our manuscript according to the reviewers’ further comments in Author- Reviewer Discussion stage.

---

### Meta-Review · Area_Chair_JqSY · 2022-09-03

**Recommendation:** Accept
**Confidence:** Certain

**Metareview:**


 This paper provides a simple method to avoid full fine-tuning of vision transformers, namely very simple linear adapters that can be trained and then subsumed into the existing linear layers during inference, which is an interesting characteristic as it prevents added computation during inference (unlike the use of regular adapters as used in NLP). Overall the reviewers appreciated the simplicity and intuition of the method, the improvement in performance over other competing methods such as VPT, the avoidance of computation overhead during inference, and comprehensive experiments.

  There were some concerns, however, related to the writing of the paper and clarity, robustness on OOD data, complexity analysis/details of runtime, and lack of theoretical justification. Many of these were addressed by the authors, including nice robustness/OOD results which add to the experimental validation. After the rebuttal, the reviewers all agreed on acceptance. While the method is still empirical (the hypothesis related to distribution matching seems unsubstantiated, and there are many other ways to do that, and so should probably not be in the paper), the paper has a strong empirical execution that uses a simple method to deal with limitations that have significant societal/deployment consequences. As a result, I recommend accepting this paper.

**Award:**

No

---

### Decision · Program_Chairs · 2022-09-14

Accept